# Comparison of Nucleic Acid Extraction Methods for a Viral Metagenomics Analysis of Respiratory Viruses

**DOI:** 10.3390/microorganisms8101539

**Published:** 2020-10-06

**Authors:** Marina Sabatier, Antonin Bal, Grégory Destras, Hadrien Regue, Grégory Quéromès, Valérie Cheynet, Bruno Lina, Claire Bardel, Karen Brengel-Pesce, Vincent Navratil, Laurence Josset

**Affiliations:** 1Laboratoire de Virologie, Institut des Agents Infectieux (IAI), Hospices Civils de Lyon, Groupement Hospitalier Nord, F-69004 Lyon, France; marina.sabatier@chu-lyon.fr (M.S.); antonin.bal@chu-lyon.fr (A.B.); gregory.destras@chu-lyon.fr (G.D.); hadrien.regue@univ-lyon1.fr (H.R.); bruno.lina@chu-lyon.fr (B.L.); 2CIRI, Centre International de Recherche en Infectiologie, Team VirPatH, Univ Lyon, Inserm, U1111, Université Claude Bernard Lyon 1, CNRS, UMR5308, ENS de Lyon, F-69007 Lyon, France; gregory.queromes@univ-lyon1.fr; 3Centre National de Référence France-Sud des Virus des Infections Respiratoires, Hospices Civils de Lyon, Groupement Hospitalier Nord, F-69004 Lyon, France; 4Laboratoire Commun de Recherche Hospices Civils de Lyon—bioMérieux, Centre Hospitalier Lyon Sud, F-69310 Pierre-Bénite, France; valerie.cheynet@biomerieux.com (V.C.); Karen.BRENGEL-PESCE@biomerieux.com (K.B.-P.); 5Université Lyon 1, Laboratoire de Biométrie et Biologie Evolutive, CNRS UMR5558, F-69100 Villeurbanne, France; claire.bardel-danjean@univ-lyon1.fr; 6PRABI, Rhône Alpes Bioinformatics Center, UCBL, Université Claude Bernard Lyon 1, F-69000 Lyon, France; vincent.navratil@univ-lyon1.fr; 7European Virus Bioinformatics Center, Leutragraben 1, D-07743 Jena, Germany

**Keywords:** viral metagenomics, next-generation sequencing, acid nucleic extraction, sample cross-contamination, kitome

## Abstract

Viral metagenomics next-generation sequencing (mNGS) is increasingly being used to characterize the human virome. The impact of viral nucleic extraction on virome profiling has been poorly studied. Here, we aimed to compare the sensitivity and sample and reagent contamination of three extraction methods used for viral mNGS: two automated platforms (eMAG; MagNA Pure 24, MP24) and the manual QIAamp Viral RNA Mini Kit (QIAamp). Clinical respiratory samples (positive for Respiratory Syncytial Virus or Herpes Simplex Virus), one mock sample (including five viruses isolated from respiratory samples), and a no-template control (NTC) were extracted and processed through an mNGS workflow. QIAamp yielded a lower proportion of viral reads for both clinical and mock samples. The sample cross-contamination was higher when using MP24, with up to 36.09% of the viral reads mapping to mock viruses in the NTC (vs. 1.53% and 1.45% for eMAG and QIAamp, respectively). The highest number of viral reads mapping to bacteriophages in the NTC was found with QIAamp, suggesting reagent contamination. Our results highlight the importance of the extraction method choice for accurate virome characterization.

## 1. Introduction

The development of metagenomics next-generation sequencing (mNGS) has enabled the exploration of whole viral nucleic acids within a clinical sample (human virome) in order to detect pathogens not targeted by conventional PCR and to identify emerging viruses [1,2,3,4,5,6,7,8]. Several studies have used various mNGS protocols to explore the human virome in diverse clinical samples, including human stools [9,10], blood [11,12], cerebrospinal fluid [13,14], human tissues [15,16], and respiratory tract samples [17,18,19,20,21]. However, the lack of standardization, the cost and duration of sequencing, and the complexity of bioinformatics analysis critically limit the wide implementation of mNGS approaches in clinical labs [22].

In particular, the extraction of viral nucleic acids is a crucial step in the molecular detection of viruses from clinical samples [23]. While there are many manual and automatic extraction methods available, it is important to choose the most sensitive and reliable one for mNGS. Numerous studies evaluating different extraction platforms in terms of their viral qPCR performance have found that the choice of extraction platform has a major impact on the reliability of the diagnostic results [24,25,26,27,28,29]. Furthermore, nucleic acid extraction methods can also impact bacteriome profiles [30,31,32] as well as the detection of particular viruses with mNGS [16,23,27,29].

Other potential issues related to extraction methods are sample cross-contamination (contamination from one sample to another) [24,33,34] and contamination by sequences present in the environment [32,34] or in the molecular biology reagents (referred to as the kitome) [35,36,37]. In viral mNGS studies, these two aspects constitute a major concern and must be precisely evaluated [38,39,40]. The impact of nucleic acid extraction methods on human virome characterization, kitome, and cross-contamination has thus far been poorly studied [41,42].

The aim of this study was to compare two automated extraction platforms commonly used in diagnostic laboratories, the eMAG (bioMérieux, Marcy-l′Étoile, France) and the MagNA Pure 24 (MP24) (Roche, Basel, Switzerland), and one manual QIAamp Viral RNA Mini Kit extraction (Qiagen, Hilden, Germany), which is among one of the most popular methods used in research laboratories. The performance of each extraction kit was evaluated in terms of (1) their ability to detect different DNA and RNA viruses in one mock sample and in clinical samples, (2) their sample cross-contamination rate, and (3) the detection of the kitome.

## 2. Materials and Methods

### 2.1. Design of the Mock Virome

The mock virome included known concentrations of five viruses isolated from respiratory samples (Table 1). These viruses were selected as representatives of a wide range of virus characteristics, such as different virion sizes (ranging from 30 to 300 nm), the presence or absence of an envelope, different genome lengths (ranging from 7 to 150 kb), different genome types (dsDNA, ssRNA), and different genome compositions (linear, segmented). All the viruses were provided by the virology laboratory at the university hospital of Lyon (Hospices Civils de Lyon). This mix contained the cell culture supernatant of Adenovirus 31 (AdV), respiratory syncytial virus A (RSV-A), herpes simplex virus 1 (HSV-1), influenza A virus, and rhinovirus. For each virus, clinical samples obtained from hospitalized patients were cultured with the appropriate cell line and media, for which the viral supernatant was then collected (Table 2).

Using the Ct values obtained by semi-quantitative real-time PCR assays (r-gene, bioMérieux, Marcy l’Étoile, France) a mix with an identical Ct value for each virus was prepared. Individual aliquots of 250 μl were prepared in triplicate for each extraction method to evaluate (9 aliquots). Aliquots were stored at −80 °C (Figure 1).

### 2.2. Sample Collection

Two additional patient samples—one positive for a DNA virus and one positive for an RNA virus—that were initially sent to our laboratory for routine viral diagnosis were also selected (Table 1). These clinical samples were a bronchoalveolar lavage (BAL) positive for HSV-1 and a nasopharyngeal aspiration (NPA) positive for RSV-A. These samples were stored at +4 °C for initial diagnosis and then diluted in transport medium (MEM medium + 1% L-glutamine + 1% fetal bovine serum + 2% Hepes) in order to obtain a sufficient volume for all the tests (up to 2.3 mL). Then, 250 μL aliquots were prepared in triplicate for each extraction method to evaluate (9 replicate samples in total). The aliquots were stored at −80°C (Figure 1).

### 2.3. Nucleic Acid Extraction

The selection of the manual kits and the different platforms was based on their commercial and hospital availabilities. The 3 different methods chosen were the NucliSENS eMAG platform (bioMérieux, Marcy l’Etoile, France), the magNA Pure 24 platform (Roche, Basel, Switzerland), and the QIAamp Viral RNA Mini Kit (Qiagen, Hilden, Germany)—all methods widely used in diagnostic laboratories. Frozen samples were thawed and homogenized by vortexing. Nucleic acids were extracted in parallel from 220 μL of the aliquot in triplicate for each kit, according to the manufacturer’s instructions. For the NucliSENS eMAG platform, specific protocol B 2.0.1 was selected. For the MP24 platform, protocol pathogen 1000 was selected. In addition, in order to evaluate the cross-contamination during automated extractions, no-template controls (NTC) were regularly interspersed between samples (i.e., 7 samples per series) (Figure 2). The QIAamp Viral RNA Mini Kit was used following the manufacturer’s recommendation with the addition of an inert Linear Acrylamide (LA) carrier (Thermo Fisher Scientific, Waltham, MA, USA) to ensure the maximum recovery of nucleic acids. To assess the reproducibility of the experimental results, the extraction and NGS analysis were set up in triplicate (for mock and respiratory samples) using the same amount of sample input.

### 2.4. Metagenomic Workflow

As previously described, we used an mNGS protocol optimized in our lab [43]. Briefly, after thawing all the samples were supplemented with MS2 bacteriophage (Levivirus genus) from a commercial kit (MS2, IC1 RNA internal control; r-gene, bioMérieux) to check the validity of the process. Only the RNA internal control MS2 was added because it validates all the steps of our protocol (including RT stage during amplification) in contrast to a DNA internal control. A no-template control (NTC) consisting of RNase free water was implemented to evaluate the contamination during the process. An additional negative control consisting of viral transport medium was added. For sample viral enrichment, a 3-step method was applied to 220 μL of vortexed sample spiked with MS2 (low-speed centrifugation, followed by the filtration of the supernatant and then Turbo DNase treatment), as described in detail in Bal et al. [43]. After viral enrichment, the total nucleic acids were extracted using one of the three methods selected for the study described above. After random nucleic acid amplification using modified whole transcriptome amplification (WTA2, Sigma-Aldrich, Darmstadt, Germany), libraries were prepared using the Nextera XT DNA Library kit and sequenced with Illumina NextSeq 500 ™ using a 2 x 150 PE high-output flow cell (Illumina, San Diego, CA, USA).

### 2.5. Bioinformatic Analysis

High-quality reads were filtered using trimmomatic PE and were further analysed using Kraken 2, followed by Braken for a taxonomic abundance estimation [44]. A custom kraken 2 database made up of (1) human, bacteria, fungi, archaea, and plasmid genome sequences given by kraken 2 and (2) an in-house viral genome database was used (viromedb, personal communication). The viromedb consists of complete viral genome sequences extracted from genbank and refseq subjected to vecscreen and seqclean softwares to remove the vectors and adaptor sequences and dustmasker to remove the low-complexity sequences.

### 2.6. Statistical Analyses

To compare the sensitivity of the three extraction methods, the mean proportion of total viral reads and specific viral abundance were determined. Kitome and sample cross-contamination were assessed by normalizing reads in reads per million (RPM), mapping the reads, and transforming them in log_10_ (RPM). For the kitome assessment, a sample was considered to be positive for a particular virus when the log_10_ (RPM) of this virus exceeded 1. Analyses were performed at the genus taxonomy level, except for the kitome, for which analyses were performed at the family taxonomy level. All the plots were constructed via ggplot2 and statistical analyses were performed with Rstatix using R (version 3.6.1). For all the statistical tests, the Student’s t-test was used.

### 2.7. Data Availability

The raw sequence data were deposited at SRA (PRJNA665071).

### 2.8. Ethics

Respiratory samples were collected for regular disease management during hospital stay and no additional samples were taken for this study. In accordance with the French legislation relating to this type a study, written informed consent from participants was not required for the use of de-identified collected clinical samples (bioethics law number 2004-800 of August 6, 2004). During their hospitalization in the Hospice Civils de Lyon (HCL), patients were made aware that their de-identified data including clinical samples may be used for research purposes, and they could opt out if they objected to the use of their data.

## 3. Results

Three different extraction methods were evaluated: two automated extraction platforms (eMAG and MP24) and a manual extraction kit (QIAamp) (Figure 1).

### 3.1. Sensitivity for the Detection of the Targeted Viruses

To evaluate the sensitivity of each method, the mean proportion of viral reads out of the total reads generated was first compared (Figure 3).

For the mock sample, both eMAG and MP24 yielded significantly higher proportions of viral reads (79.9% and 84.5%, respectively) in comparison with QIAamp (59.4%; *p* < 0.05 and *p* < 0.01, respectively). For the HSV-positive BAL, eMAG yielded a higher proportion of viral reads (2.6%) compared with QIAamp (0.5%, *p* < 0.01), but did not significantly differ with the MP24 (3.9% viral reads). For the RSV-positive NPA, a similar trend was observed, with an average of 4.3%, 3%, and 1.4% viral reads for eMAG, MP24, and QIAamp, respectively.

To determine potential bias in the detection of DNA or RNA viruses, the relative abundance of *Levivirus* (Internal Quality Control) and targeted viruses in each triplicate were then compared (Figure 4).

*Levivirus* was detected in all samples for MP24 and QIAamp, and in 8/9 samples for eMAG. For the targeted viruses, all the viruses were detected with all the extraction methods (Figure 4).

For the mock sample, a difference in the relative abundance of both the RNA and DNA viruses was noted when comparing the extraction methods.

The highest relative abundance of RNA viruses was observed using the QIAamp method (8.2% *Enterovirus*, 2.2% *Alphainfluenzavirus,* and 0.6% *Orthopneumovirus*), which was not significantly different from that of eMAG (5% *Enterovirus*, 1.4% *Alphainfluenzavirus,* and 0.4% *Orthopneumovirus*). The lowest relative abundance was obtained with MP24 (2.5% *Enterovirus*, 0.4% *Alphainfluenzavirus,* and 0.2% *Orthopneumovirus*; *p* < 0.05 only for *Alphainfluenzavirus and Orthopneumovirus* compared to the QIAamp method).

For DNA viruses, the highest relative abundance was obtained using eMAG and MP24 (54.4% and 69.3% for *Simplexvirus*, respectively, vs. 23.4% with QIAamp; 9.5% and 17.1% for *Adenovirus*, respectively, vs. 1.3% with QIAamp; *p* < 0.01).

Surprisingly, many reads associated with *Dependoparvovirus*, an ssDNA virus, were observed in the mock sample after the QIAamp extraction (29.4% with eMAG, 9.9% with MP24, and 61.4% with the QIAamp extraction).

### 3.2. Sample Cross-Contamination

The impact of the different extraction methods on sample cross-contamination was then evaluated from the NTC samples included between each sample during the extractions (Figure 2) by mapping the read count of viruses that were present in samples from the same batch: internal quality control (MS2, *Levivirus*) and targeted viruses (*Adenovirus*, *Orthopneumovirus, Simplexvirus*, *Alphainfluenzavirus*, *Enterovirus*, and *Dependoparvovirus*) (Figure 5).

*Levivirus* was not found in any NTC extracted by eMAG and QIAamp but was found in 1/9 NTCs extracted by MP24 (MS2 log_10_RPM = 2.1).

For the two automatic extractors, the main contaminant was *Simplexvirus* (HSV1), found in 4/9 NTCs (up to 1.5 log_10_(RPM)) and in 5/9 NTCs (up to 3.7 log_10_(RPM)) extracted with eMAG and MP24, respectively. For eMAG, there was also a high level of *Dependoparvovirus* contamination in 1/9 NTCs (log_10_(RPM) = 1.2) and of *Orthopneumovirus* in 1/9 NTCs (log_10_(RPM) = 2.1). For MP24, cross-contamination was noted in the NTC, with all viruses contained in the mock 4/9 NTCs with *Mastadenovirus* (up to 2.9 log_10_(RPM)), 5/9 NTCs with *Dependoparvovirus* (up to 2.8 log_10_(RPM)), 3/9 NTCs with *Enterovirus* (up to 2.1 log_10_(RPM)), 3/9 NTCs with *Alphainfluenzavirus* (up to 2.1 log_10_(RPM)), and 3/9 NTCs with *Orthopneumovirus* (up to 1.3 log_10_(RPM)).

For the QIAamp manual extractor, the main contaminant was *Alphainfluenzavirus* (3/3 NTC, up to 2.2 log_10_(RPM)). There were also contaminations with *Mastadenovirus* (2/3 NTC, up to 1.5 log_10_(RPM)) and with *Enterovirus* (2/3 NTC, up to 1.8 log_10_(RPM)).

Overall, the viral sample cross-contamination represented on average 0.002%, 0.107%, and 0.015% of the total reads generated from the NTC for the eMAG, MP24, and QIAamp extraction methods, respectively (corresponding to 1.53%, 36.09%, and 1.45% of the viral reads for the eMAG, MP24, and QIAamp extraction methods, respectively).

Importantly, for MP24 we also noted that sample cross-contamination was more associated with a batch effect than with the position of the sample in the extraction cartridge. Hence, while NTC#1, #2, and #3 were all contaminated, there was less contamination in all NTCs from batch #2 than in the NTCs from the two other batches of extractions (Figure 5).

### 3.3. Kitome Assessment

The impact of the different extraction methods on the viral kitome contamination was then evaluated by detecting in the NTC and TM the presence of reads associated with viruses other than the targeted viruses. The log_10_(RPM) of the kitome was significantly higher with the QIAamp extraction compared to the eMAG and MP24 extractions (*p* < 0.01).

The viral kitome contamination generated from the NTC represented an average of 11.31 log_10_(RPM), 16.88 log_10_(RPM), and 70.77 log_10_(RPM) with the eMAG, MP24, and QIAamp extraction methods, respectively (Figure 6a).

A total of 19, 28, and 55 different viral families were detected in the NTC with the eMAG, MP24, and QIAamp methods, respectively (Figure 7a).

The contaminants derived mainly from bacteriophage families. In particular, *Siphoviridae* was found in all three methods (ranging from 2.32 to 3.39 log_10_(RPM)), corresponding to 21.4%, 13.7%, and 4.8% of the total viral kitome reported with the eMAG, MP24, and QIAamp extraction methods, respectively.

Regarding the transport medium, the viral kitome contamination represented an average of 24.11 log_10_(RPM), 19.94 log_10_(RPM), and 72.45 log_10_(RPM) with the eMAG, MP24, and QIAamp extraction methods, respectively (Figure 6b). The same main viral families associated with kitome were found for the two automatic extractors with a majority of *Poxviridae*, while for the manual extractor the main family found was *Siphoviridae* (Figure 7b).

Overall, the kitome contamination was higher with the QIAamp extraction (*p* < 0.01). *Siphoviridae* bacteriophages were found in the three methods, while other contaminants such as *Poxviridae* were specifically found in the transport medium extracted by the automated methods.

## 4. Discussion

In this study, we compared the performance of three extraction methods commonly used in clinical laboratories for viral mNGS analysis (eMAG, MagNA Pure 24, and QIAamp Viral RNA Mini Kit). The extractors yielding the highest proportion of viral reads were the two automatic extractors, eMAG and MP24. A previous study found that Qiagen kits tend to extract a high proportion of human nucleic acids, which could explain the lower viral proportion reported in the present study [29].

Despite this difference, all the viruses present in the mock or clinical samples could be detected with all the methods evaluated herein. Nonetheless, a difference in the relative abundance of RNA and DNA viruses was noted in the mock sample. The highest relative abundance of RNA viruses was reported with QIAamp, and the highest relative abundance of DNA viruses was reported with the eMAG and MP24 platforms. This bias should be taken into account during the interpretation of mNGS studies and underlines the importance of the extraction method choice, depending on the virus to be explored.

These differences could be due to the various properties of viruses, including the presence of an envelope, the type of genome, or the size of the virions. Yang et al. showed a better performance of RNA virus recovery with EasyMag (identical silica extraction technology and similar performance to that of Emag [45]) compared to the MagNA Pure Compact. They explained this difference by a possible RNA degradation or by an ineffective binding of RNA to the magnetic beads [30]. Finally, the higher sensitivity of the QIAamp kit for the detection of RNA viruses might be explained by the kit having been initially intended for the extraction of RNA viruses. The detection of DNA viruses would still be possible through the capture of the RNA transcripts of DNA viruses. Meanwhile, a higher number of reads on RNA viruses for the QIAMP method was noted from the mock sample including both RNA and DNA viruses at equi-Ct; the supplementation of both DNA and RNA internal controls in NTCs would have been interesting for evaluating also the extraction bias in low-biomass samples. In addition to the differences related to the extraction methods, certain types of viruses or genomes can be preferentially amplified as described for the *Poliovirus* by Lewandoska et al. [23].

Moreover, bioinformatics analysis of mNGS data can also impact the viral reads annotation. The presence of gaps at the level of taxonomic classification (genus) can bias the interpretation for these given taxonomic groups (notably phages). The choice of the viral reference database is therefore crucial in order to limit viral misclassifications or the lack of detection of new emerging viruses [46]. Moreover, short reads reduce the accuracy of viral read assignation. As there is currently no gold standard in de novo assembly software for virome assessment, extensive benchmarking will be necessary in order to choose the most adapted method for future studies. With the new advances in third-generation sequencing and the improvement in sequence quality generated by this approach, we expect that the use of longer reads will led to an increase in specificity as compared to short-read technology.

Interestingly, we observed many reads associated with *Dependoparvovirus* only in the mock sample, especially with the QIAamp extraction. This can be explained by the presence of ADV in the mock, which might be associated with *Adeno-associated dependoparvovirus* [47] or with contaminants from the QIAamp column [35,48]. Our internal control (MS2, *Levivirus*) was detected for all replicates extracted with the three methods, except for two extracts with eMAG. As previously described, competition between the target viruses and MS2 might be observed during the process, leading to undetected MS2 [43].

To date, few studies have assessed the impact of different extraction methods on the performance of metagenomics. Klenner et al. evaluated four manual QIAamp nucleic acid extraction kits (QIAamp Viral RNA Mini Kit, QIAamp DNA Blood Mini Kit, QIAamp cador Pathogen Mini Kit, and QIAamp MinElute Virus Spin Kit) with four different viruses (including *Reovirus*, *Orthomyxovirus*, *Orthopoxvirus*, and *Paramyxovirus*), and reported that the selection of the kits has only a minor impact on the yield of viral reads and the quantity of reads obtained by NGS [28]. However, this study only evaluated manual kits from the same manufacturer and with separate RNA and DNA extraction methods.

Conversely, several studies have highlighted the importance of the nucleic acid extraction protocol in producing high-quality extracts suitable for sequencing [23,27,29]. Lewandoska et al. compared the impact of three extraction methods (QIAamp Viral RNA mini Kit, PureLink Viral RNA/DNA Mini Kit, and automated NucliSENS EasyMAG) on the recovery of different viral genomes (*adenovirus*, *poliovirus*, HHV-4, *influenza A virus*). The EasyMAG extraction was more efficient for both RNA and DNA viruses, leading to a higher recovery of viral genomes. The mNGS results are highly susceptible to inaccurate conclusions resulting from the sequencing of contaminants [36]. In the present study, the viral contamination was higher with MP24 than with eMAG and QIAamp. Automated extraction platforms may lead to sample cross-contamination due to the generation of aerosols or robotic errors. Knepp et al. compared two automated extractors (BioRobot M48 instrument (Qiagen, Inc.) and MagNA Pure) and did not show contamination with the automated instruments [24]. However, they only investigated cross-contamination related to *enterovirus* (RNA virus), unlike our study, which evaluated a panel of RNA and DNA viruses.

The second source of contamination may come from the reagents (kitome) used throughout the process or from laboratory contaminants. The extractor for which the kitome abundance was highest was the QIAamp. The main contaminants were *Siphoviridae*, *Myoviridae*, *Microviridae*, and *Podoviridae*, which is consistent with other studies that have reported similar findings with spin columns [34]. In order to monitor the kitome and avoid misinterpretation, it is important to implement negative controls at different steps of the process [38,42,49]. Here, we did not add any internal controls in the NTC in order to get the highest sensitivity in detecting the kitome (without using reads to sequence MS2). On the other hand, the internal control MS2 was added in the TMs and was detected in all except one batch of the eMAG extraction in order to estimate the potential contaminants present in the transport medium, as it was previously published that fetal bovine serum contains DNA [50,51].

Furthermore, a computational approach for removing contaminants of viral origin should be developed, as previously described for bacteriome data [34,52].

Although the results show higher sample cross-contamination with the MP24 and higher kitome-related contamination with the QIAamp, other steps throughout sample processing can produce contamination. Here, we did not include an NTC at each step of the process to control for other sources of contamination. In addition, only a few respiratory samples were tested herein, and so further studies on a larger number of respiratory samples, as well as on other types of samples (stool, blood, and tissue) or other respiratory viruses (such as the SARS-CoV-2, which has the largest human RNA virus genome), should be performed. While three commonly used extraction methods have been evaluated in this study, it could be interesting to test others.

## 5. Conclusions

Our findings highlight the importance of extraction method choice for viral mNGS analysis. The eMAG platform yielded a higher proportion of viral reads, with a limited impact of reagents and sample cross-contamination compared to the QIAamp and MP24 extractors.

## Figures and Tables

**Figure 1 microorganisms-08-01539-f001:**
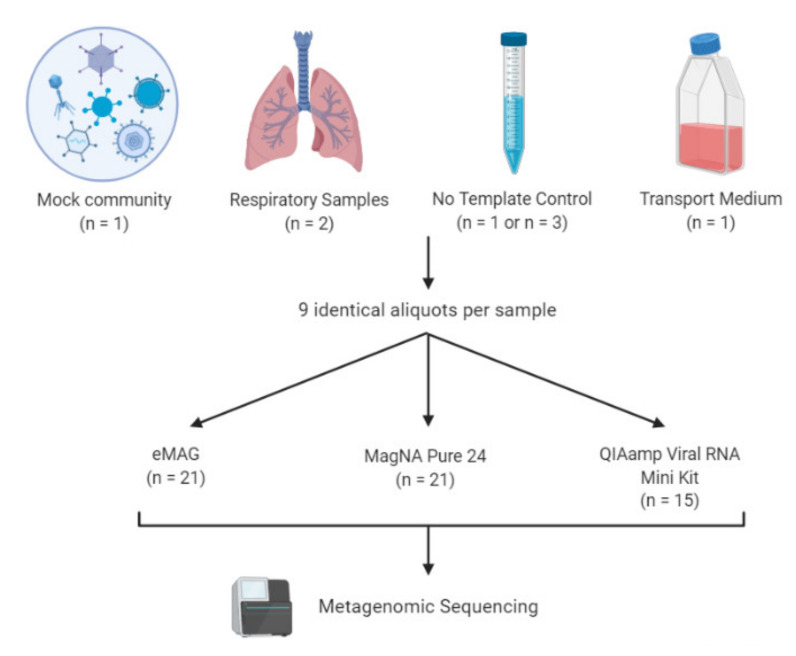
Overview of the study design. A mock virome containing five viruses isolated from respiratory samples representative of a wide range of virus characteristics (adenovirus 31, respiratory syncytial virus A, herpes simplex virus 1, influenza A virus, and rhinovirus) was prepared. Human clinical samples were obtained from hospitalized patients (one bronchoalveolar lavage positive for herpes simplex virus 1 (HSV-1) and one nasopharyngeal aspiration positive for respiratory syncytial virus A (RSV-A). No template controls (NTCs) and transport medium samples were implemented in the process to control for sample and kitome cross-contamination. For the automatic extractors, NTCs were interspersed between the samples of each batch (n=3 per batch), whereas for the manual method only 1 NTC was included per batch (n = 1). To assess the reliability and reproducibility of the experimental results, the extractions and next generation sequencing (NGS) workflow were set up in triplicate (for the mock and respiratory samples), using the same amount of sample input. Finally, libraries were sequenced in the same run with Illumina NextSeq 500 ™ using a 2 x 150 paired-end (PE) high-output flow cell.

**Figure 2 microorganisms-08-01539-f002:**
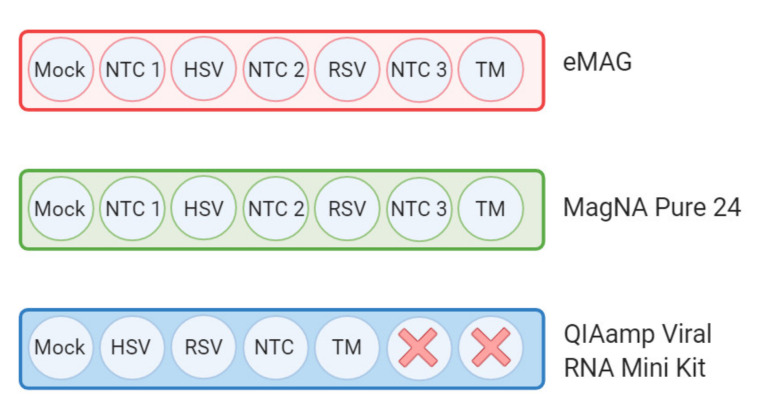
Arrangement of the samples on the extraction platforms. The three different extraction methods used here were the NucliSENS eMAG platform (bioMérieux, Marcy l’Etoile, France), MagNA Pure 24 platform (Roche, Basel, Switzerland), and manual QIAamp Viral RNA Mini Kit (Qiagen, Hilden, Germany). Nucleic acids were extracted in triplicate from the same aliquot for each kit according to the manufacturer’s instructions. In order to evaluate cross-contamination during the automated extractions, an NTC was regularly interspersed between samples (7 samples per series). To evaluate the kitome contamination, a transport medium sample was added in addition to NTC. Here, each color represents a different extraction method. NTC: No Template Control; HSV: Herpes Simplex Virus; RSV: Respiratory Syncytial Virus; TM: Transport Medium.

**Figure 3 microorganisms-08-01539-f003:**
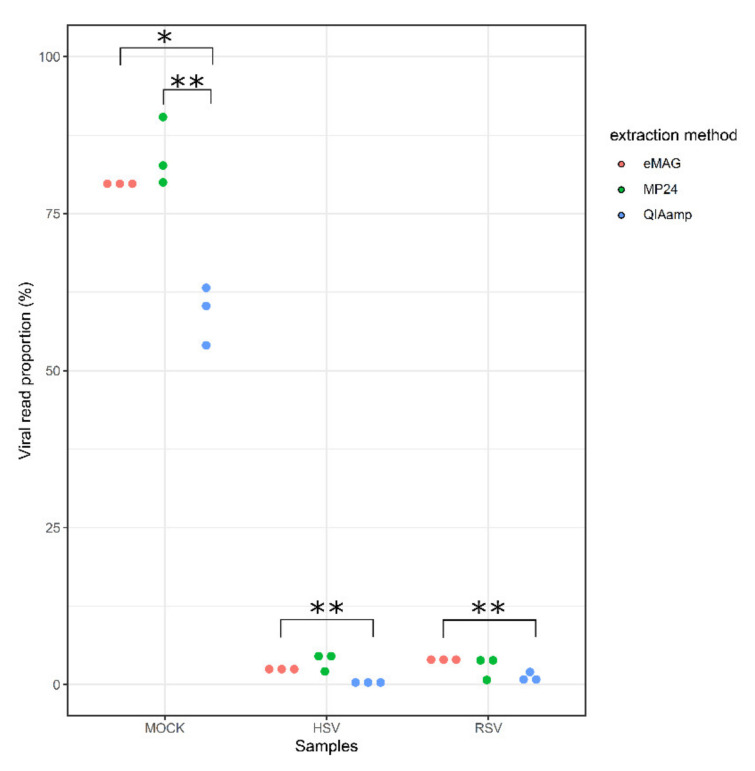
Distribution of the viral read proportion (%) according to the three extraction methods. QIAamp Viral RNA Mini Kit (QIAamp, blue dots), MagNA Pure 24 (MP24, green dots), and eMAG (eMAG, red dots) extraction methods for mock and clinical samples (HSV and RSV). HSV: herpes simplex virus; RSV: respiratory syncytial virus. The average of viral read proportions was compared two by two using a Student’s t test (**p* < 0.05 and ***p* < 0.01).

**Figure 4 microorganisms-08-01539-f004:**
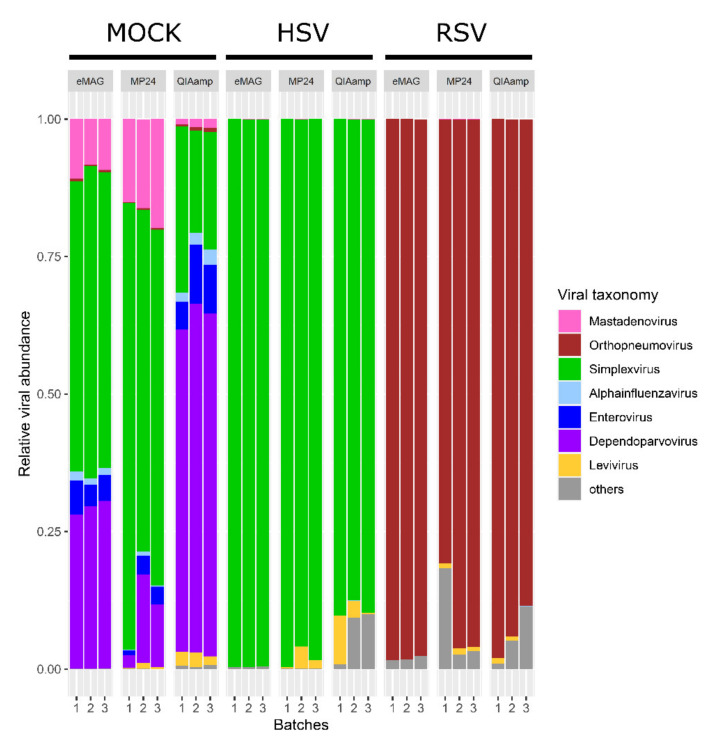
Taxonomic distribution (relative abundance) of triplicates for mock and clinical samples according to the three extraction methods. The relative distribution is described at the genus taxonomic level. Only major viral sequences are illustrated with different colours. HSV: herpes simplex virus; RSV: respiratory syncytial virus.

**Figure 5 microorganisms-08-01539-f005:**
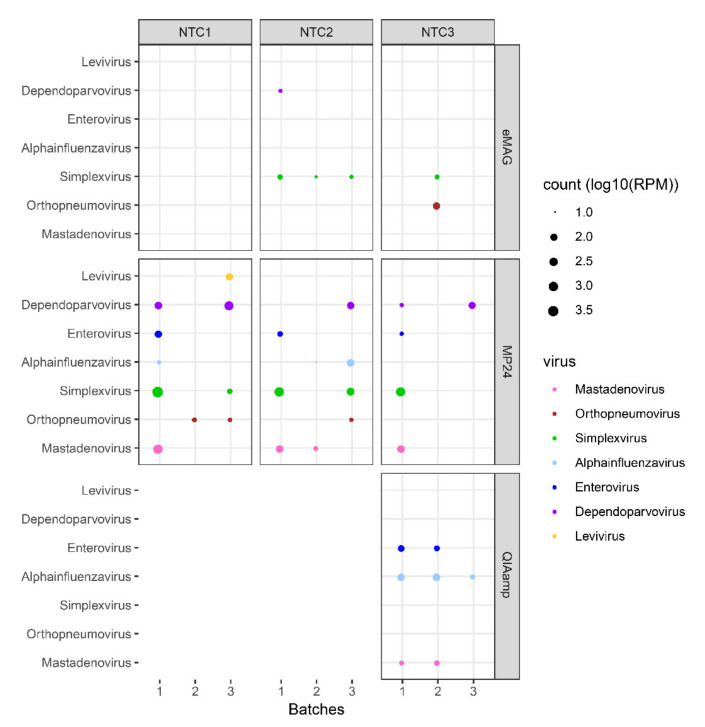
Sample cross-contamination. Bubble plot showing the normalized abundance in log_10_(RPM) of the targeted viruses in each NTC and for the different extraction methods. During the automated extractions, NTCs were interspersed between samples (i.e., 3 NTCs per batch). For manual extraction, only one NTC was added. Analyses were performed at the genus level. Each genus is represented by coloured dots. The size of the dots represents the abundance normalized in log_10_(RPM) for viral reads associated with sample cross-contamination. HSV: herpes simplex virus; RSV: respiratory syncytial virus; NTC: no template control; RPM: reads per million.

**Figure 6 microorganisms-08-01539-f006:**
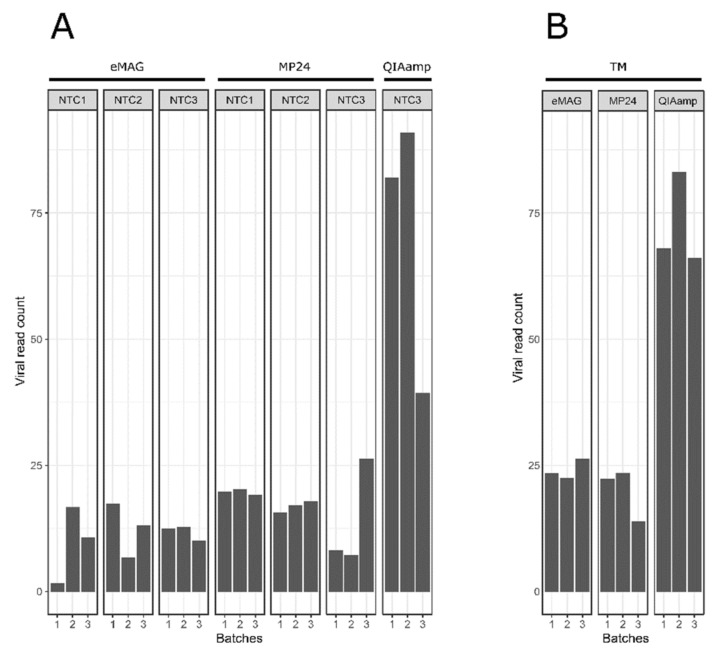
Proportion of Kitome contained in each triplicate from the different extraction methods (**A**) in NTC and (**B**) in the transport medium. Bar plot showing the sum of the viral read count normalized in log_10_(RPM), associated with reagent contamination (i.e., reads associated with other viruses than the targeted viruses: kitome) for each NTC and TM compared between different extraction methods. During automated extractions, NTCs were interspersed between samples (i.e., 3 NTCs per batch). For manual extractions, only one NTC was added. In addition to the NTC, a transport medium was added. Analyses were performed at the family taxonomy level. A sample was considered to be positive for a particular virus when the log_10_(RPM) of this virus exceeded 1. HSV: herpes simplex virus; RSV: respiratory syncytial virus; NTC: no template control; TM: transport medium; RPM: reads per million.

**Figure 7 microorganisms-08-01539-f007:**
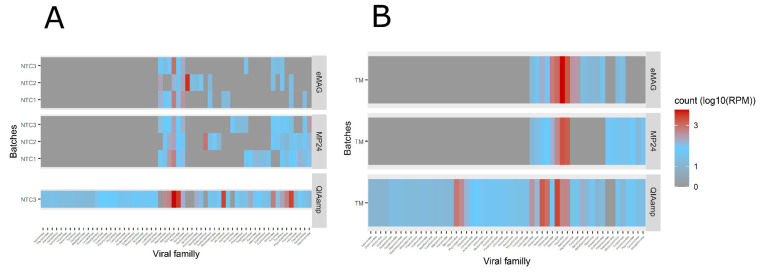
Virus family presence contained (**A**) in NTC and (**B**) in TM (other than the target virus) in each triplicate from the different extraction methods. The heatmap showed the viral family read count associated with the kitome (the presence of reads associated with other viruses than the targeted viruses) normalized in log_10_(RPM) in each NTC and TM between the different extraction methods. A gradient of colors was defined from gray (no count) to blue (few counts) to red (highest counts). During the automated extractions, the NTCs were interspersed between samples (i.e., 3 NTCs per batch). For manual extraction, only one NTC was added. In addition to the NTC, transport medium was added. Analyses were performed on the family taxonomical level. A sample was considered to be positive for a particular virus when the log_10_(RPM) of this virus exceeded 1. HSV: herpes simplex virus; RSV: respiratory syncytial virus; NTC: no template control; TM: transport medium; RPM: reads per million.

**Table 1 microorganisms-08-01539-t001:** List of the five selected viruses included in the mock virome and the clinical respiratory samples. The mock virome consists of known concentrations of five viruses isolated from respiratory samples, selected as representatives of a wide range of virus characteristics (virion size, the presence or absence of an envelope, genome length, genome type (dsDNA, ssRNA), and genome composition (linear, segmented)). All the viruses were provided by the virology laboratory (Hospices Civils de Lyon). Human clinical respiratory samples were obtained from hospitalized patients. dsDNA: double stranded DNA; ssRNA: single stranded RNA.

Samples	Virus	Virus Family	Molecular Typing	Baltimore Classification	Genome Composition	Genome Size (kb)	Virion Size (nm)	Enveloped	Ct Value
	Adenovirus	*Adenoviridae*	ADV-A31	Group I: dsDNA	Linear	34	65/80	No	21.1
	Respiratory Syncytial Virus	*Paramyxoviridae*	RSV-A	Group V: ssRNA (-)	Linear	15	150	Yes	22.1
Mock Virome	Herpes Simplex Virus	*Herpesviridae*	HSV-1	Group I: dsDNA	Linear	150	120/300	Yes	26.2
	Influenza Virus	*Orthomyxoviridae*	IAV	Group V: ssRNA (-)	Segmented	13	80/120	Yes	20.4
	Rhinovirus	*Picornaviridae*	HRV-A13	Group IV: ssRNA (+)	Linear	7	30	No	29
Clinical samples	Herpes Simplex Virus	*Herpesviridae*	HSV-1	Group I: dsDNA	Linear	150	120/300	Yes	16.6
	Respiratory Syncytial Virus	*Paramyxoviridae*	RSV	Group V: ssRNA (-)	Linear	15.2	150	Yes	19.3

**Table 2 microorganisms-08-01539-t002:** Cell culture and media. The different types of cells used for culture are HEp-2 cells (human liver cancer cells, ATCC CCL-23), Vero cells (monkey kidney epithelial cells, ATCC CCL-81), MRC-5 cells (human fetal lung fibroblasts, Biowhittaker, 25-10-1995 produced by RD-Biotech (Besançon, France)), and MDCK cells (canine kidney epithelial cells, ATCC CCL-34). FBS: foetal bovine serum; HEp-2: human epithelial cell line type 2; MDCK: Madin-Darby canine kidney; MEM: minimal essential medium.

Virus	Cells	Nature of the Sample	Culture Media	Additional Elements	Number of Days in Culture	Cryoprotectant Medium
Adenovirus	Hep	stool	MEM	2% penicillin-streptomycin + 1% L-glutamine + 0.05% neomycin + 2% FBS + 2% Hepes Buffer	4	Yes
Respiratory Syncytial Virus	Hep	nasal throat	MEM	2% penicillin-streptomycin + 1% L-glutamine + 0.05% neomycin + 2% FBS + 2% Hepes Buffer	4	Yes
Herpes Simplex Virus	Vero	vaginal swab	MEM199	2% penicillin-streptomycin	3	No
Influenza A	MDCK	bronchoalveolar lavage	MEM	2% penicillin-streptomycin + 1% L-glutamine + 0.05% neomycin + 0.05% Trypsine + 2% Hepes Buffer	5	Yes
Rhinovirus	MRC5	tracheobronchial aspiration	MEM	2% penicillin-streptomycin + 1% L-glutamine + 0.05% neomycin + 2% FBS + 2% Hepes Buffer	5	Yes

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
