# Peer review of "Comparison of Nucleic Acid Extraction Methods for a Viral Metagenomics Analysis of Respiratory Viruses"

_microorganisms, 2020, doi:10.3390/microorganisms8101539_

Round 1

Reviewer 1 Report

In this manuscript, the authors compared three nucleic acid extraction methods for metagenomics viral detection and conducted some bioinformatics analyses. The experimental design and the resulting outcomes are plausible.

My suggestions for improvement are as follows

  • Viral enrichment was mentioned in metagenomics workflow, but no detail was shown. Please descript.
  • Both DNA and RNA internal controls were used, but only IC1 RNA from MS2 phage was mentioned in Materials and Methods, not IC2 DNA from Levivirus. Please add in.
  • Based upon the results, I assume the internal controls were not added into NTC or TM. It would have been better to add these additional controls in initial experimental design, so as to reduce any potential bias. Notably, NGS is sensitive in detection, especially when Nextera XT requires 1 ng input of DNA. NTC and TM are supposed to contain no DNA.
  • When short-reads are used directly for Kraken2, it shows high background or bias due to some short-reads from NGS as well as some mis-classifications. It certainly reduces such background and bias by using kraken2 after short-reads assembly.  
  • It should be noted that QIAamp viral RNA mini kit is mainly targeting RNA extraction, including ssDNA, although dsDNA is extracted as well; whereas eMAG and MP24 are targeting both DNA and RNA extraction in automation. Such differences should have been demonstrated better if both DNA and RNA internal controls added into NTC and TM as additional controls.
  • Bias caused by NGS and bioinformatics analysis could be discussed.
  • Minor: Line 142, “acid nucleic” should be “nucleic acid”.

Reviewer 2 Report

I have a minor but essential comment. While it is a fact that HSV-1 can cause respiratory symptoms, it is not classified as a respiratory virus. This should be changed.

In addition, it would also be of interest to test some more extraction methods beyond the three included but which are also frequently used. Moreover, I wonder why no coronavirus was included, which these days could be more important than ever before.
